# Biomineralization of amorphous Fe-, Mn- and Si-rich mineral phases by cyanobacteria under oxic and alkaline conditions

Karim Benzerara[1]*, Agnes Elmaleh[1], Maria Ciobanu[2], Alexis De Wever[1], Paola Bertolino[2], Miguel Iniesto[2], Didier Jézéquel[3], Purificación López-García[2], Nicolas Menguy[1], Elodie Muller[1], Fériel Skouri-Panet[1], Sufal Swaraj[4], Rosaluz Tavera[5], Christophe Thomazo[6,7], David Moreira[2]

[1]Sorbonne Université, Muséum National d'Histoire Naturelle, UMR CNRS 7590. Institut de Minéralogie, de Physique des Matériaux et de Cosmochimie (IMPMC), Paris, France
[2]Unité d'Ecologie Systématique et Evolution, CNRS, Université Paris-Saclay, AgroParisTech, Orsay, France
[3]IPGP, CNRS UMR 7154, Université de Paris & UMR CARRTEL, INRAE-USMB, France

[4]Synchrotron SOLEIL, L'Orme des Merisiers, Saint-Aubin-BP 48, 91192 Gif-sur-Yvette, cedex, France

[5]Departamento de Ecología y Recursos Naturales, Universidad Nacional Autónoma de Mexico, DF Mexico, Mexico

[6]Biogéosciences, CNRS UMR 6282, Université de Bourgogne Franche-Comté, France

[7]Institut Universitaire de France, Paris, France

**Correspondence to*: Karim Benzerara (karim.benzerara@sorbonne-universite.fr)

*Keywords: Biomineralization; Iron; Manganese; Cyanobacteria*

**Abstract.** Iron and manganese are poorly soluble elements in oxic and alkaline solutions, whereas they are much more soluble under anoxic conditions. As a result, the formation of authigenic mineral phases rich in Fe and/or Mn has traditionally been viewed as diagnostic of global or local anoxic conditions. Here we reveal that some specific cyanobacteria of very small size (<2 µm, *i.e.* picocyanobacteria) can biomineralize abundant, authigenic Fe(III)-, Mn(IV)- and Si-rich amorphous phases under oxic conditions in an alkaline lake in Mexico. The resulting biominerals cluster as small globules arranged as rings around the division septum of cyanobacterial cells. These rings are enveloped within an organic, likely polysaccharidic envelope, and are partially preserved, at least morphologically, upon sedimentation. Based on their 16S rDNA sequence, these cyanobacteria were affiliated to the Synechococcales order. The high Fe and Mn enrichment of the biominerals questions the systematic inference of anoxic conditions based on their detection. Moreover, this process scavenges iron from the water column, an overlooked biological contribution to the Fe cycle. Finally, it reveals a new case of controlled biomineralization of Si-rich phases by bacteria.

## 1 Introduction

Iron is the fourth most abundant element in the Earth's crust (Taylor, 1964). It is essential to a broad diversity of organisms, particularly primary producers (Liu et al., 2021). However, Fe is poorly soluble in oxic, neutral to alkaline waters and sometimes becomes a limiting nutrient for the biosphere (Tagliabue et al., 2017). Overall, its bioavailability affects the efficiency of atmospheric carbon uptake in aqueous environments (Aumont and Bopp, 2006). Therefore, efforts have been invested to model the Fe cycle in these environments (Boyd and Ellwood, 2010). Fe enters the cycle through different sources such as aeolian dust, detrital particles, release from anoxic sediments, or hydrothermal vents (Jickells et al., 2005; Elrod et al., 2004; Tagliabue et al., 2010). Biological activity sets a "ferrous wheel" by mediating the dissolution of particulate Fe and degrading biogenic Fe-organic complexes on the one hand, and through Fe cell uptake and complexation by organic ligands on the other hand. Adsorption on various settling mineral particles can scavenge iron out of this wheel. Overall, the balance between removal and release controls the bioavailability of Fe. In any case, since dissolved Fe concentrations remain low under these conditions, no significant authigenesis of Fe-rich mineral phases is expected in an oxic water column. By contrast, soluble reduced Fe(II) was much more abundant in ancient aqueous environments before atmosphere oxygenation, 2.4 billion years ago (Canfield, 2005). At that time, authigenesis of Fe-rich mineral phases was usual, either by precipitation of Fe(II) phases and/or local oxidation of Fe(II) in Fe(III), resulting in the massive deposition of sedimentary Fe-rich formations such as banded iron formations (BIFs) (Canfield, 2005). The nature of the authigenic minerals feeding these formations has been widely discussed. Among the possible candidates, Fe(II)-rich silicates such as greenalite have been suggested (Rasmussen et al., 2019). Like iron, reduced Mn(II) is highly soluble, while oxidized Mn (III, IV) precipitates as insoluble oxides under oxic conditions (Johnson, 2019). Consequently, the finding of Fe- and/or Mn-rich authigenic minerals has often been considered as an indication of past anoxic conditions (Rasmussen et al., 2019; Tosca et al., 2016).

Over the last years, we have studied several crater lakes in Mexico with a main interest for the formation of carbonate microbialites harboured by these lakes (Zeyen et al., 2021). In some of the shore-based microbialites, we found that local seepages of anoxic groundwater could feed the instant precipitation of Fe-containing phases, either as Fe-Mg layered double hydroxides in Lake Alchichica, where dissolved orthosilicic acid concentrations are low, or Fe-bearing kerolite/stevensite in other lakes, where dissolved $H_4SiO_4$ concentrations are higher (Zeyen et al., 2019). To further characterize the Fe and Mn cycles in these lakes, we also systematically examined the particulate fraction in their water column, using a combination of approaches at the bulk and the micro- to nano-scale. This produced one fortuitous find based on the examination of the oxic water column of Lake La Preciosa: we evidenced a biological process forming a Fe- and Mn-rich mineral phase under oxic and alkaline conditions, a process of particular interest considering the aforementioned considerations. In the following, we characterize the structure and chemical composition of these biominerals and their association with cell structures, demonstrate

their biogenicity, and identify the microorganisms biomineralizing them. This allows discussing the environmental and biological importance of this possibly overlooked biomineralization process.

## 2 Materials and Methods

### 2.1 Field campaign and sample collection

Cyanobacteria forming (Fe,Mn)-rich phases were detected in all field campaigns to Lake La Preciosa at all depths (0 to 35 m), in May 2014, May 2016, March 2018 and May 2019. Temperature, dissolved oxygen concentration, turbidity and pH were measured in situ over a depth profile using a YSI 6600 multiparameter probe and an optode $O_2$ SDOT nke in May 2016. The detection limit for dissolved $O_2$ was ca 0.1 % (0.3 ± 0.2 µM). The oxygen sensor was calibrated against water-vapor-saturated ambient air (100 % $O_2$ local saturation) and the "zero" $O_2$ was checked using a sodium sulphite solution at 10 wt%. The pH sensor was calibrated using 3 buffers (4/7/10). Lake water was sampled with a Niskin bottle in the center of the lake at different depths from the surface down to 35 m, the maximum depth of Lake La Preciosa. Part of the water was filtered through 0.2 µm polyethersulfone (PES) filters on the same day and kept in sterile tubes for chemical analyses. Part of filtered solutions were acidified with nitric acid (2%) for major cation analyses. The other part was used for measurement of anion and orthosilicic acid ($H_4SiO_4$) concentrations, without pre-acidification. Lastly, microbial cells were concentrated 5000 times from 5 L unfiltered water samples using 0.2-µm pore-size Cell-Trap units (MEM-TEQ Ventures Ltd, Wigan, UK).

### 2.2 Fluorescence-activated cell sorting (FACS) and cell micromanipulation

Cyanobacterial cells were sorted with a BD FACSAria III Cell Sorter (Becton Dickinson and Company, San Jose, California, USA) equipped with one gas-state 633 nm laser (>18 mW, elliptical shape) and three solid state lasers at 488 nm (>20 mW, elliptical shape), 405 nm (>50 mW, elliptical shape), and 375 nm (> 7mW, top hat shape). The 488 nm laser was used for the analysis of forward scatter (FSC, 488/10, 1.0 ND filter), side scatter (SSC, 488/10), phycoerythrin (PE, 585/42, 556LP) and chlorophyll/phycocyanin (PerCP-Cy5-5, 695/40, 655LP). Light was detected by Hamamatsu R3896 photomultiplier tubes in C10562 sockets (Hamamatsu, 211 Hamamatsu City, Japan). The applied voltages were: FSC (220 V), SSC (324 V), PE (703 V) and PerCP_Cy5-5 (577 V). The trigger was set on the FSC with a threshold of 2000. The fluidic system was run at 45 psi (3.102 bar) with an 85-µm nozzle. Samples were sorted at a speed of 6000-6500 events. $s^{-1}$, with a flow rate of 1, corresponding to 0.2 psi over the sheath pressure. The sheath fluid consisted of sterile 1X PBS buffer ($K_2HPO_4$ 0.11%, $KH_2PO_4$ 0.03%, NaCl 0.8%). In addition to the excitation lasers, an additional BD Accudrop laser of 660 nm was used for Drop Delay setup. Lake La Preciosa cell suspensions were filtered on a 35 µm mesh just before sorting to avoid clogs. We used 1 µm UV Fluoresbrite

Microspheres 24062 (Polysciences, Warrington, USA) for calibration of the log range and a pure culture of *Synechococcus rubescens* as a positive control for phycoerythrin and chlorophyll/phycocyanin fluorescence (Wood et al., 1985). Twenty-five thousand events of one distinct phycoerythrin/chlorophyll-positive population (Pop1, Supplementary Fig. 6a-b) were sorted directly on a 13 mm, 0.4 µm pore size polycarbonate filters (MerckMillipore, Germany) using the most accurate sorting mode "Single-cell purity". The filters were prepared and observed by SEM as described below. For DNA extraction, 20 cells were sorted under the same conditions, directly into a 0.2 mL microtube (Eppendorf) and stored at -20°C. Cytometric data were processed using the BD FACSDIVA V9.0.1 software (Becton, Dickinson and Company, San Jose, California, USA).

We also isolated cyanobacterial cells by micromanipulation from samples collected in March 2018 using an Eppendorf PatchMan NP2 micromanipulator equipped with 6 µm-diameter microcapillaries (Eppendorf) mounted on a Leica Dlll3000 B inverted microscope. Individual cells were rinsed twice with sterile 10 mM Tris pH 8.0 buffer. Several sets of 5 to 10 cells were deposited in a volume of 0.4 µL of the same buffer in 0.2 mL tubes (Eppendorf) and stored frozen at -20°C until further processing.

### 2.3 16S rRNA gene sequencing and phylogenetic analysis

DNA was extracted from the FACS-sorted and the micromanipulated cells with the PicoPure DNA extraction kit (Applied Biosystems). In the case of the micromanipulated cells, whole genome amplification (WGA) was carried out on the PicoPure-extracted DNA using Multiple Displacement Amplification (MDA) with the REPLI-g WGA kit (Qiagen). 16S rRNA genes were amplified by PCR from the PicoPure-purified and WGA-amplified DNAs using the cyanobacterial-specific primers CYA-106F (5′-CGG ACG GGT GAG TAA CGC GTG A) and CYA-1380R (5′-ACG ACT TCG GGC GTG ACC). PCR amplifications were carried out in 25 µl reaction volume with GoTaq polymerase reaction mix (Promega, Lyon, France). PCR reactions consisted of an initial denaturing period (95°C for 3 min) followed by 35 cycles of denaturing (93°C for 45 s), annealing (55°C for 45 s), extension (72°C for 2 min), and a final extension period (72°C for 5 min). Because of the possible presence of more than one species in the samples, we constructed 16S rRNA gene clone libraries with the PCR amplicons using the Topo TA cloning system (Invitrogen) following the instructions provided by the manufacturer. After plating, positive transformants were screened by PCR amplification using the M13R and T7 flanking vector primers. Amplicons were sequenced by Sanger sequencing (Genewiz, Essex, UK).

All forward and reverse Sanger sequences were quality-controlled and merged using Codon Code Aligner (http://www.codoncode.com/aligner/). Then, we utilised Mafft (Katoh and Standley, 2013) to produce multiple sequence alignments including our sequences and the closest blast (Altschul et al., 1997) hits identified in GenBank. Gaps and ambiguously aligned regions were removed with TrimAl (Capella-Gutiérrez et al., 2009). A maximum likelihood phylogenetic tree was constructed with IQ-TREE (Nguyen et al., 2015) using the general time reversible (GTR) model and branch support estimated by ultrafast bootstrapping.

**2.4 Sediment collection**

A sediment core measuring 10 cm in length was collected from the bottom of Lake La Preciosa at 35-m depth during the field campaign in May 2016, using a gravitational Uwitec corer with a diameter of 90 mm. The sediment core was transferred into a glove-bag and placed under anoxic conditions ($N_2$ atmosphere) immediately after collection. It was then processed and sliced on the field into cm-scale fractions along the core's vertical axis. Oxygen levels in the glove-bag were monitored with an Oxi 340i WTW oxygen meter and were always below the detection limit of 0.1 mg/L. Back in France, sediments were vacuum-dried in a Jacomex™ glovebox (< 1 ppm $O_2$) before being analyzed by SEM.

**2.5 Solution chemistry**

Orthosilicic acid ($H_4SiO_4$) concentrations were determined by continuous flow colorimetric analyses (QuAAtro Axflow) at the Institut de Physique du Globe de Paris (IPGP, Paris, France) using non-acidified, 0.2-µm-filtered La Preciosa solutions. Anion concentrations were measured using ion chromatography (ICS1100 Dionex Thermofisher) using non-acidified, 0.2-µm-filtered La Preciosa solutions. Concentrations of major cations were determined by inductively coupled plasma - optical emission spectrometry (ICP-OES, iCAP6200 Thermofisher), using acidified, 0.2-µm-filtered La Preciosa solutions. The uncertainty in the concentration measurements of orthosilicic acid, anions and cations was lower than 5%. Activities of anions, cations and orthosilicic acid as well as saturation indices of the surface water solutions of the lakes were calculated using the Visual MINTEQ software.

**2.6 Confocal laser scanning microscopy (CLSM) and correlative with scanning electron microscopy (SEM)**

For standard CLSM analyses, cells concentrated using Cell-Trap units were deposited on a glass slide, covered by a coverslip and sealed with nail polish. Samples were observed by CLSM using a Zeiss LSM 710. Excitation was performed at 405 and 488 nm and emission spectra were measured for each pixel of the images over the 405-720 wavelength range using a 34-channel Quasar T-PMT detector. Data were processed using the Zeiss Zen software. For correlative CLSM-SEM, cells were first washed with milliQ water and then deposited on a coverslip and left to dry. The coverslip was mounted onto a KorrMik Life Sciences sample holder and the correlative Shuttle and Find software implemented in ZEN 2012 was used. CLSM analyses were conducted first. The same areas were relocated in the SEM with the Shuttle and Find software.

**2.7 Scanning electron microscopy (SEM)**

Two types of sample preparation were conducted for scanning electron microscopy (SEM) analyses: (i) in one case, cells were sorted by FACS directly on a 13 mm, 0.4 µm pore size polycarbonate membrane (MerckMillipore, Germany), washed with 1

mL sterile milli-Q water and dried at room temperature; (ii) in another case, cell suspensions were vitrified by plunge and freeze in liquid ethane on a gold planchet before being freeze-dried in a Leica EM ACE600 apparatus. Observations were performed using a Zeiss ultra 55 field emission gun SEM. The acquisition of high spatial resolution images was performed at an accelerating voltage of 1 kV and a working distance of ~3 mm and a 20 nm aperture using an annular in-column InLens detector for detecting secondary electrons. For the search of Fe- and Mn-rich rings in samples, backscattered electron (BSE) images were acquired using an angle selective backscattered (AsB) detector at an accelerating voltage of 10 kV and a working distance of ~7.5 mm and a 60 nm aperture at high current. This also allowed to count cells over large mosaics (e.g., 730*550 $\mu m^2$), converting numbers into cell density (cells. $mL^{-1}$) based on the ratio of the total filter surface to the surface of the mosaic, divided by the volume of filtered water in mL. To estimate the error bar, the mosaic was divided into 4 quadrants to achieve 4 counting replicates and derive a standard deviation. The elemental composition of mineral phases was determined under the same conditions as BSE imaging by energy dispersive x-ray spectrometry (EDXS) using an EDS QUANTAX detector (Bruker). EDXS data were analyzed using the ESPRIT software package (Bruker).

### 2.8 Transmission electron microscopy (TEM), scanning transmission electron microscopy (STEM), and energy filtered-TEM (ETFEM)

Cells concentrated from the water column using Cell Trap units were washed before being deposited on a formvar-coated TEM grid and air-dried. Fe- and Mn-rich rings were more fastidious to find in sediments because of the high contrast of all mineral particles composing them. Therefore, we enriched samples in rings by a weak acid leaching (step addition of acetic acid keeping pH at ~6). These samples were also deposited on a formvar-coated TEM grid for further analyses.

Observations of unfixed samples were carried out using a Jeol 2100F TEM microscope operating at 200 kV, equipped with a Schottky emitter, a STEM device, which allows Z-contrast imaging in the high-angular annular dark-field (HAADF) mode, a Jeol Si(Li) x-ray detector and a GIF 2001 Gatan energy filter. Semi-quantitative analyses of EDXS spectra were performed using the JEOL Analysis Station software. This was based on the use of K-factors. After subtracting out the background noise in the EDXS spectrum, the software performed a Gaussian fit of selected elemental peaks and calculated the area under the peaks. From this, the atomic percentage of each selected element was assessed. EFTEM elemental mapping of C, Fe and Mn was performed using the 3-window technique (Hofer et al., 1997). This technique requires three energy-filtered images: two positioned before the ionisation edge (pre-edge images), which serve to calculate the background, and one positioned just after the edge (post-edge image). Calculated background image was subtracted from the post-edge image to give an elemental map, in which changes in background shape were considered. Maps were calculated for the C K-edge and Fe and Mn L2,3 edges using a 20-eV-wide (for C) or 40-eV-wide (for Fe, Mn) energy window for pre-edge and post-edge. Zero-loss images were obtained by selecting elastically scattered electrons only, using a 10-eV-wide energy window.

## 2.9 Scanning transmission x-ray microscopy (STXM)

Cells suspended in the water column and sediment samples were prepared in the same way as for TEM and deposited on TEM grids. Scanning transmission x-ray microscopy analyses were performed on the HERMES beamline at the SOLEIL synchrotron (St. Aubin, France). This microscopy uses monochromated x-rays in the soft x-ray energy domain (200-2000 eV) which are focused onto a ~25 nm spot by a Fresnel zone plate. This approach provides images with a ~25 nm spatial resolution and spatially-resolved speciation information based on x-ray absorption near edge structure (XANES) spectroscopy. An image is obtained by positioning the sample at the focal point of the lens and raster scanning it in x and y, while recording the intensity of the transmitted x-rays. Some cells with rings were pre-located by short TEM imaging conducted prior to STXM measurements in order to facilitate the analyses. However, in order to assess potential electron beam damages, we also analyzed by STXM cells in areas of the TEM grids kept pristine, with no prior TEM analyses. A stack of STXM images of the areas of interest was acquired at a sequence of photon energies at the C K-edge first. The spectral resolution was 0.12 eV in the 282–291.5 eV energy range, where most of the narrowest XANES peaks were present. Then, stacks were acquired at the Mn L2,3-edges (spectral resolution of 0.1 eV in the 642-650 and 653.75-662 eV ranges) before Fe L2,3-edges (spectral resolution of 0.15 eV in the 709-721 eV range). Areas free of particles were used to measure the incident flux (I0). Images were converted from transmitted intensity units to optical density units (OD) following the formula OD= –log(I/I0). XANES spectra were extracted and mapped using the aXis2000 software (McMaster University, http://unicorn.mcmaster.ca/axis/aXis2000-IDLVM.html).

# 3 Results

## 3.1 A new biomineralization process of amorphous (Fe,Mn, Si)-rich phases

We sampled water at several depths in the Mexican crater Lake La Preciosa, up to ~40 m below water level and examined the plankton (>0.2 µm). Among the broad morphological diversity of microorganisms observed by light microscopy, one frequent morphotype appeared as doublets of coccoid cells measuring ~1 µm in diameter each and forming loose aggregates of up to ~50 cells (Supplementary Fig. 1). Their abundance was estimated to be about 2.3 (±0.3) x $10^5$ cells/mL. An opaque band systematically separated each doublet of dividing cells. This trait morphologically resembles that of previously described members of the *Cyanocatena* genus (Hindák, 1982). With electron microscopy, these bands appeared as bright rings with a diameter of 1 µm (Fig. 1-2; Supplementary Fig. 2). They localized at the division septum of cells, and were sometimes observed in planar view when detached from dividing cells (Fig. 1 b-d & 2a). Overall, these rings appear as traces of septation events.

In some cases, we observed single rings orientated perpendicularly in-between 2 pairs of dividing cells (Supplementary Fig. 1). This pattern suggests that cells divided along 2 consecutive perpendicular planes. By contrast, in rare cases, we observed a series of aligned cells with rings perpendicular to the alignment, suggesting a single division direction only.

At higher magnification, the rings appeared as aggregates of globules measuring ~50-100 nm in diameter (Fig. 2b; Supplementary Fig. 2) and presenting a nanoporous, sometimes fibrous texture (Supplementary Fig. 2T, V). The rings mostly contained Fe (~33.6 mol% on average) and Si (~35.9 mol% on average), with some variable amounts of Mn (av. 6.5 mol%) with lower amounts of Mg, Ca and K as shown by energy dispersive x-ray spectrometry (EDXS) (Table SI-1; Fig. 2 g-l & Supplementary Fig. 2). They systematically had a low Al content (1.02 ±1.38 mol%, 25 measurements; Table SI-1). The (Fe+Mn+Ca+Mg)/(Si+Al) ratio was estimated at 1.5±0.55 on average and varies between 0.8 up to 2 (25 measurements). Electron diffraction revealed that the grains comprising the rings were amorphous (Fig. 2c). No Fe- and/or Mn-rich intracellular deposits were detected within the cells. The redox state of Mn and Fe in the rings was determined by x-ray absorption near edge structure (XANES) spectroscopy using scanning transmission x-ray microscopy (STXM). Analyses at the Fe and Mn $L_{2,3}$ edges of pristine rings unirradiated by TEM beforehand showed that Fe and Mn composing the rings were oxidized, *i.e.,* in the Fe(III) and Mn(IV) redox states (Supplementary Fig. 3). By contrast, Mn composing rings irradiated by TEM at low magnification beforehand was systematically reduced. This demonstrates that TEM induces artefactual reduction of Mn in these phases.

### 3.2 Bacteria biomineralizing (Fe, Mn)-rich rings are cyanobacteria phylogenetically close to *Cyanobium*

The coccoid cells forming the (Fe, Mn, Si)-rich rings showed strong autofluorescence by confocal laser scanning microscopy (CLSM), with two maximum emission peaks at ~585 and 670 nm (Supplementary Fig. 4). Correlation of CLSM with SEM confirmed that these autofluorescent cells were the ones forming (Fe, Mn, Si)-rich rings (Supplementary Fig. 5). Based on Wood *et al*. (Wood et al., 1985) and a direct comparison with the autofluorescence of reference phycoerythrin-containing *Synechococcus rubescens* cells, these two peaks were interpreted as resulting from the fluorescence of phycoerythrin and chlorophyll/phycocyanin, respectively. This indicated that cells forming (Fe,Mn)-rich rings are cyanobacteria.

These phycoerythrin-containing cyanobacteria were efficiently enriched by fluorescence-activated cell sorting (FACS) based on the autofluorescence properties of their particular pigments (Supplementary Fig. 6a-b). DNA was extracted from the enriched cell fractions and, after PCR amplification, their 16S rRNA genes were sequenced. In parallel, micromanipulated cells were also used for 16S rRNA gene sequencing after whole genome amplification (WGA). Maximum likelihood phylogenetic tree reconstruction confirmed that these cells were cyanobacteria. More specifically, their sequences branched within a group containing sequences classified as *Synechococcus* and *Cyanobium* (Fig. 3). The small differences between La Preciosa sequences (<1%), suggested a population of very closely related cyanobacterial strains producing (Fe, Mn)-rich rings in this lake.

These 16S rRNA gene sequences were used to screen a large dataset of 16S rRNA gene metabarcoding sequences of plankton from 11 Mexican lakes, including La Preciosa (Zeyen et al., 2021; Iniesto et al., 2022). We detected sequences closely related (>99% identity) to those of the (Fe, Mn, Si)ring-producing cyanobacteria in nine of these eleven lakes: La Preciosa, Alchichica, Atexcac, Aljojuca, Alberca de los Espinos, Pátzcuaro, Quechulac, Tecuitlapa and Yuriria (Supplementary Table 2). These sequences were particularly abundant (representing >1% of all bacterial metabarcoding reads) in La Preciosa, Alchichica, Atexcac, Aljojuca, and Alberca de los Espinos, sometimes reaching frequencies up to 15%. Interestingly, these lakes were characterized by relatively high salinity and alkalinity values (Zeyen et al., 2021), whereas those where the sequences were completely absent had the lowest values for these parameters.

### 3.3 (Fe, Mn)-rich rings form under oxygenated conditions and within a polysaccharidic extracellular compartment

Although we have not experimentally determined the speciation of Fe in La Preciosa water column, we can infer that the concentration of dissolved free Fe is likely below 1 nM (Liu and Millero, 2002) based on the high pH (between 8.8 and 9.0) and oxic conditions (>25% of local $O_2$ water saturation), in particular in the upper part of the water column (Supplementary Fig. 7). The rest of Fe in the water column of Lake La Preciosa may be (i) colloidal, inorganic, or organic (complexed by dissolved organic molecules) Fe, in the <0.2 μm dissolved fraction, or (ii) particulate Fe complexed by microbial cells and/or adsorbed/co-precipitated in inorganic particles. Measured dissolved Fe concentrations (*i.e.* free and colloidal Fe) were always below the detection limit of ICP-AES (0.02 nM), except at a depth of 5 m in 2019, when Fe and Mn were anomalously high,

reaching 0.1 and 1 μM respectively. For that date, dissolved Mn concentration was below the detection limit as well at other depths. Here, Fe and Mn were most likely under a colloidal form, suggesting that the size of the colloidal pool varies over time.

Dissolved Si concentrations were measured and found to be relatively high in Lake La Preciosa ($5x10^{-4}$ M). Moreover, as observed in several other alkaline Mexican lakes (Zeyen et al., 2021), Lake La Preciosa solutions were oversaturated with amorphous sepiolite (Wollast et al., 1968), a hydrated Mg-silicate phase, at all depths and all times (Supplementary Fig. 7).

Lastly, chemical conditions prevailing locally where the rings form may be different from the ones in the lake. Indeed, we found evidence of an extracellular compartment enclosing the rings. First, SEM observations at low-electron voltage of lyophilized samples showed that rings were embedded in a fibrillar matrix, with a mesh texture reminiscent of the extracellular polymer substances (EPS) excreted by some bacteria (Fig. 1e-f). Second, energy-filtered TEM (EFTEM) imaging showed that the globular grains were contained within a carbon-rich envelope (Supplementary Fig. 8). Finally, the detection of a carboxylic-rich polymer using XANES spectroscopy at the C K-edge confirmed the polysaccharidic composition of this carbonaceous envelope, based on the presence of a major absorption peak at 288.6 eV, which is interpreted as $1s\rightarrow \pi^*$ electron transition in carboxylic C and has often been used as a spectroscopic marker for acidic polysaccharides (Benzerara et al., 2004) (Supplementary Fig. 8).

**3.4 Biomineralized cyanobacterial rings are morphologically preserved upon sedimentation**

The sediments were mostly composed of aragonite, anorthite and diatomaceous amorphous silica (Supplementary Fig. 9). Moreover, in the bulk Fourier transform infrared spectroscopy spectra of the sediments, bands at 473, 540, 1026, 3625 and 3680 $cm^{-1}$ could be affiliated to clay-like phases, a proportion of which possibly corresponded to the rings. Although they could not be unambiguously recognized by bulk analyses, cyanobacterial rings with a well preserved, sometimes partly broken morphology, were abundant enough so that they could be observed by electron microscopy in sediment samples with no prior chemical treatment (Fig. 4 and Supplementary Fig. 10-12). TEM-EDXS showed that similar to rings in the water-column, they had a relatively high Fe content (Fe/Si~0.36 atom/atom vs ~1 in the water column). However, sediment rings had a different chemical composition compared to those found in the water column with a relatively higher Mg content (~19 mol% in sediments vs. 3.8 mol% on average in the water column) and a lower Mn content (~1.5 mol% vs 6.5 mol% in the water column). However, we note that this statement is based on few analysed sediment rings only and will need further support from additional analyses, with a particular attention to some potential progressive transformation within the water column and

with depth in the sediments. The surface appearance of some rings suggested that they were contained within an organic envelope (Fig. 4a and Supplementary Fig. 10 a), which was further supported by STXM analyses at the C K-edge on acetic-acid-leached samples (Supplementary Fig. 4). Here, the spectral signature of carbon was different from that measured on water-column rings. However, this may have resulted from artifactual damages induced by the acid-leaching of the samples, TEM pre-screening and/or transformation of the organics upon sedimentation. Overall, bulk chemical analyses of the sediments showed that $Fe_2O_3$ represented up to 2% and MnO 0.1% of the total sediment mass (including ignition loss). This provides some maximum estimate of the Fe- and/or Mn-containing ring abundance.

## 4 Discussion

### 4.1 Biological and environmental significance of this biomineralization process

Our findings show that some picocyanobacteria biomineralize Fe-, Si- and Mn-rich phases in oxic and alkaline water. The intensity and environmental distribution of this biomineralization process remain unknown. Several groups of picocyanobacteria described using classical approaches seem to form Fe-rich precipitates as well, with diverse textural patterns (Hindák, 2002). For example, *Cyanodictyon imperfectum*, detected in several lakes worldwide, forms rings composed of iron oxides at its division septum (Cronberg and Weibull, 1981; Economou-Amilli and Spartinou, 1991). *Cyanocatena planctonica* has also been described to form rings composed of iron oxides at the division septum (Hindák, 1982). All these cells divide along one division plane only, a taxonomic feature different from most Lake La Preciosa cyanobacteria. Hindak (Hindák, 1982) described another cyanobacterium, *Cyanogranis ferruginea*, which divides along 2 planes and forms iron oxide rings but not at the cell septa, in contrast with the cells in Lake La Preciosa. No Mn or Si enrichment was reported by any of these studies but mineralogical analyses were too limited to be conclusive. One difficulty in inferring phylogenetic affiliations is that sequence data for all these cyanobacterial morphotypes is not available. Although the morphological features of picocyanobacteria alone cannot be used to reliably discriminate between different taxa (Callieri et al., 2012), the morphotype observed in La Preciosa resemble *Cyanocatena* cells (Hindák, 1982), at least regarding their biomineralization pattern. Therefore, we propose to ascribe our new 16S rDNA sequence to this genus, as *Cyanocatena* sp. The taxonomic affiliation of this sequence allows to start investigating the potential prevalence of this unusual biomineralization process in different lakes by the analysis of massive metabarcoding datasets. While the enrichment in Fe and Mn is moderate in Lake La Preciosa sediments, higher sedimentary enrichments be discovered using such an approach.

Although we ignore the molecular mechanisms of this biomineralization process (see section 4.2), several potential functions can be speculated for this trait: (i) High amounts of Fe and Mn are needed by cyanobacteria compared with other microorganisms (Raven et al., 1999; Nelson and Junge, 2015) but, at the same time, elevated intracellular concentrations of these elements may enhance oxidative stress, therefore requiring some appropriate homeostasis (Liu et al., 2021; Kranzler et al., 2013). Different systems are involved in bacterial intracellular storage of Fe, including different types of ferritin family

proteins (Keren et al., 2004). The sequestration of high amounts of Fe and Mn by extracellular amorphous phases may contribute to homeostasis and provide a larger but less toxic Fe and Mn reserve (Brown et al., 2005; Cosmidis and Benzerara, 2022). (ii) Alternatively, Lingappa et al. (2021) showed high intracellular accumulation of Mn in some cyanobacteria and suggested that it could be used as a catalytic antioxidant, useful in environments with high oxidative stress. Here Mn may instead be accumulated outside the cells and serve similar purposes. (iii) The oxidation of Fe and Mn upon extracellular release may also generate protons which could be used to generate a proton gradient and gain energy, a mechanism suggested for Fe-oxidizing bacteria (Chan et al., 2004). (iv) Last, the sequestration of high amounts of Fe and/or Mn in these biominerals may be a way to divert these elements from other organisms and therefore serve competition purposes, as suggested for the production of siderophores by some cyanobacteria in the case of iron stress (Wilhelm, 1995).

### 4.2 A new controlled biomineralization process concentrating Fe and Mn under oxic conditions

The chemical composition of the biomineralized rings (Al-free and a (Fe+Mn+Ca+Mg)/(Si+Al) ratio between 0.8 and 2) is reminiscent of the composition of various phyllosilicate phases such as smectites [ferrosaponite: Fe/Si~0.75] and/or serpentines [greenalite or cronstedtite: Fe/Si=1.5 to 4]. Alternatively, the lowest measured (Fe+Mg+Mn)/(Si+Al) values may correspond to Fe-bearing talcs [(Mg+Fe)/Si=0.75], consistently with phases detected in La Preciosa sediments and Mexican microbialites (Zeyen et al., 2019) or a mixture of amorphous Fe and Mn oxyhydroxides sorbing silica. The exact nature of the local structure of this phase could be determined in the future using techniques such as extended X-ray absorption fine structure (EXAFS) spectroscopy at, e.g., the Si and Fe K-edges, providing the rings can be separated from other Fe-, Mn- and Si-phases in the samples. In any case, considering its chemical composition, this biomineralized phase may serve as a precursor phase to silicates. The initial phase may consist of mostly Fe(II)- and Mn(II)-silicates that subsequently oxidized, since decades of experimental syntheses have consistently shown that the synthesis of Fe(III)-smectites was achieved by starting from an Fe(II)-silicate gel (Baron et al., 2016). Alternatively, the amorphous (Fe, Mn, Si)-rich phase may form by the release of Fe(II) and Mn(II) at the septum of the bacterial cells, which would spontaneously oxidize in the orthosilicate-rich water of Lake La Preciosa. This scenario is consistent with the formation of Fe-Si complexes with an Fe/Si ratio between 1 and 2 observed in several experimental studies (Pokrovski et al., 2003; Doelsch et al., 2001). Here, the high Si content of the rings may just be related to the Si-rich environmental conditions and not the result of bacterial Si cycling. Several studies have documented the extracellular neoformation of diverse clay-like phases by bacteria (Konhauser and Urrutia, 1999; del Buey et al., 2021). They do so by binding anions (e.g., $H_3SiO_4^-$) to positively-charged molecular sites on the cell surface or extra-polymeric substances (EPS), and by ion-bridging with cations bound to negatively charged functional groups and/or by increasing the local pH due to their metabolic activity (Zeyen et al., 2015). Here, while these mechanisms may also occur owing to the EPS envelope surrounding the rings and the possible pH increase due to oxygenic photosynthesis which favours the precipitation of hydroxides and silicates (e.g., Zeyen et al., 2015), the control on biomineralization by this specific bacterium appears much

tighter. First, this biomineralization process is not seen in other cyanobacteria populating Lake La Preciosa and therefore is likely mediated by some biological mechanisms specific to these cyanobacteria. Moreover, (i) the biominerals have a specific ring morphology, with a definite diameter of ~1 μm; (ii) they are associated with the cell division plane, (iii) they are enclosed within an extracellular EPS compartment that might be involved in mineral precipitation and control the unusual textural organization of the (Fe, Mn, Si)-mineral phase- as a ring, and (iv) they are rich in Fe and Mn, whereas the concentration of free Fe under these environmental conditions does not allow abiotic authigenesis of Fe-rich phases. This shows that these cyanobacteria manage to efficiently concentrate Fe and Mn locally from the water column, thus likely involving some energy cost. At this stage, the specific mechanisms involved in this process can only be speculated. Fe complexed by organic molecules is usually the most bioavailable form for cyanobacteria (Beghoura et al., 2019). Alternatively, Fe contained in colloids and particles may be released into the water column through biologically and photochemically-mediated dissolution (Baker and Croot, 2010). The uptake of Fe from one or several of these sources might be achieved based on processes similar to those evidenced in many other cyanobacteria, such as reductive uptake (Kranzler et al., 2014) and/or siderophore-mediated Fe(III) uptake (Swanner et al., 2015) but possibly with higher efficiency by these cyanobacteria. Mn is usually taken up in its reduced Mn(II) state through the same transporters as Fe (Qiu et al., 2021). Moreover, some studies have reported significant Mn stores within some cyanobacterial cells (Lingappa et al., 2021). The possibility that Mn in the rings is derived from such an internal store should be investigated. However, here, no intracellular precipitate rich in Mn and/or Fe was detected within the cyanobacteria, suggesting that Fe(II) and Mn(II) are rapidly released after uptake at the division septum where they precipitate to form rings. Whether Fe(II) and/or Mn(II) may be specifically released during cell division remains to be investigated. Moreover, additional lines of inquiry are to determine (i) whether proton uptake by the cells could occur locally at the division septum, locally increasing pH and favouring the precipitation of these phases, and (ii) how the EPS envelope around the rings form or preform before the precipitation of the Fe- and Mn-rich rings. The isolation of this bacterium in cultures and/or the analysis of its genome should help in the future to answer these different questions.

## 5 Conclusions

Whatever its mechanism and function, the biomineralization of (Fe, Mn)-rich rings appears as a yet overlooked Fe-scavenging process, which, together with the already known Fe-scavenging formation of lithogenic inorganic particles, removes Fe out of the water column. Interestingly, Fe-Mn nodules and crusts on the seafloor are present under oxic conditions too and their formation may involve microorganisms as well (Hein and Koschinsky, 2014). However, nodules have been suggested to be diagenetic in origin, involving microbially catalysed oxidation of Mn sourced by reduced fluids, whereas the formation of the

rings observed in the present study do not likely require the input of reduced Fe(II) and Mn(II) to the bacteria. Cyanobacteria have been considered as actors in the formation of some iron deposits by triggering the oxidation of dissolved Fe(II) from anoxic fluids and therefore inducing the precipitation of Fe-rich mineral phases (Crowe et al., 2008; Emerson and Moyer, 2002). The sheaths of some cyanobacteria can also serve as precipitated iron repository in iron-depositing hot springs (Brown et al., 2005). Here, we evidence an additional mechanism by which cyanobacteria may contribute to the formation of Fe-rich phases, involving Fe scavenging and directed nucleation and growth of Fe-precipitates within an organic matrix. The resulting biominerals show diagnostic features, which may help search for them in modern and ancient samples, following the example of the recent discovery of purported cyanobacteria microfossils with intracellular Fe-silicate nanocrystals in the 1.88 Ga Gunflint Iron Formation (Lepot et al., 2017). The embedding of Fe-rich rings within EPS in La Preciosa may enhance their preservation and, concomitantly, the formation of these organominerals may favour the preservation of organic matter upon time, as shown by Keil et al. (Keil et al., 1994). Consistently, and despite some chemical transformations suggested by the differences in chemical compositions between sediment and water column rings, at least part of these rings withstand dissolution and remain morphologically preserved in Lake La Preciosa sediments, having therefore some potential for fossilization over longer time periods. Overall, the existence of such a biomineralization process questions the systematic inference of anoxic conditions based on the detection of such Fe- and Mn-rich phases in the sedimentary record.

**Data availability**

**Author contributions**

KB, AE, MC, PLG, RT, CT, DM designed the study. KB, AdW, PB, MI, DJ, PLG, RT, CT, DM collected the samples on the field. KB, AE, MC, AdW, PB, MI, DJ, PLG, NM, EM, FSP, SS, RT, DM carried out the measurements. KB, AE, MC, ADW, MI, NM, EM, DM analyzed the data. All authors wrote the manuscript.

**Competing Interests**

The authors declare that they have no conflict of interest.

**Acknowledgements**

This study was supported by the INSU programme Interrvie, the ERC grants Calcyan (PI: K. Benzerara, Grant Agreement no. 307110) and Plast-Evol (PI: D. Moreira, Grant Agreement no. 787904), and the Agence Nationale de la Recherche (PI: P. López-García, ANR Microbialites, grant number ANR-18-CE02-0013-02). We thank Lucie Franco and Michael Bourges for helping with FACS experiments; Mélanie Poinsot, Jena Johnson, Daniel Nothaft for help in the field; Franck Bourdelle for

help on STXM experiments; Mickael Trichet for help with lyophilisation. The authors acknowledge SOLEIL for the provision of beamtime.

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

**Figures**

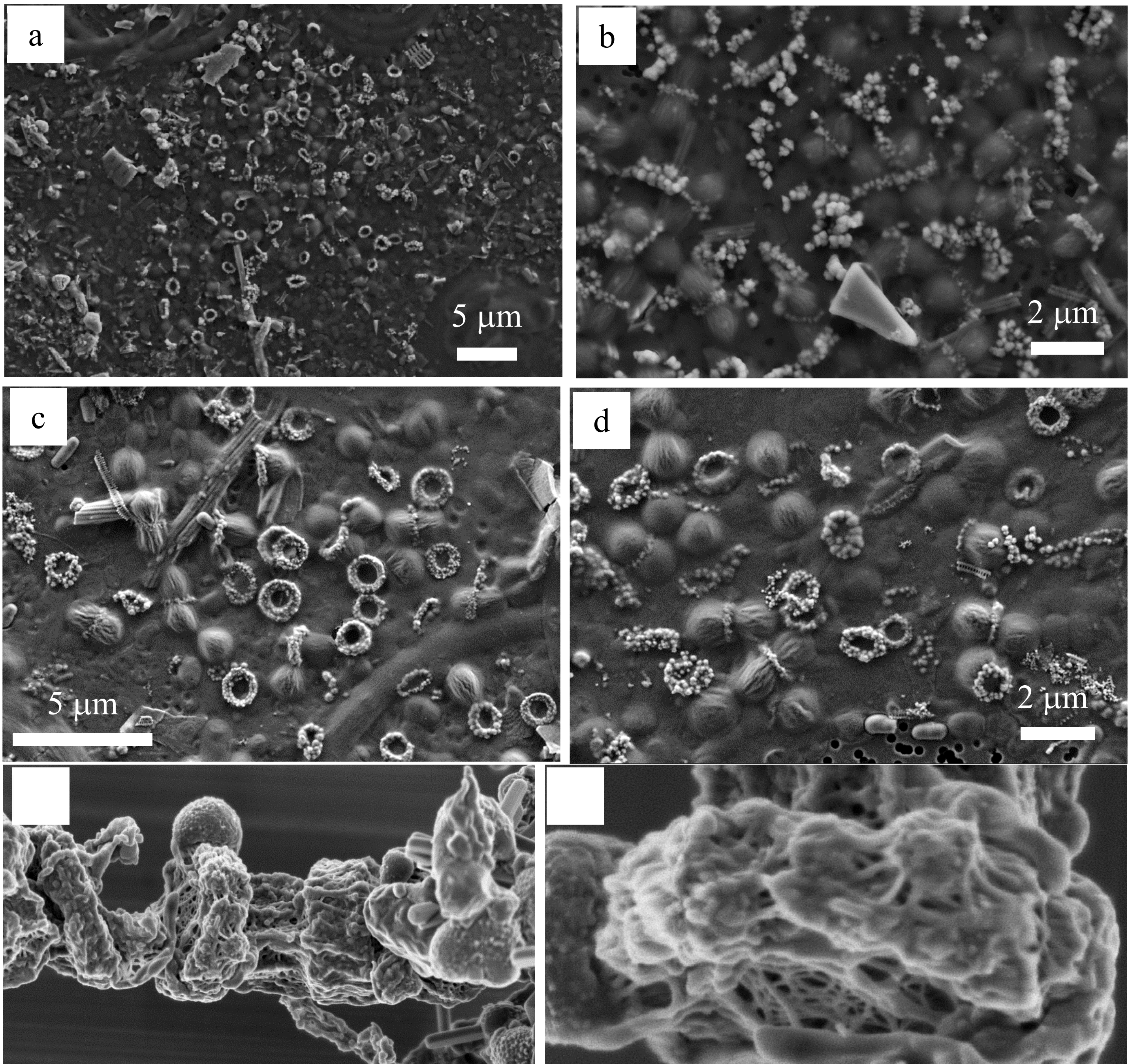

**Figure 1.** Scanning electron microscopy imaging of (Fe,Mn, Si)-rich rings formed by dividing cells. (a-b): SEM images obtained with the secondary electron detector of an air-dried sample collected in 2018 (acceleration voltage: 5 keV; working distance: 7 mm). (c-d): SEM images obtained with the secondary electron detector of air-dried samples collected in 2019 (acceleration voltage: 2 keV; working distance: 3 mm). (e-f): SEM images obtained with the InLens detector of freeze-dried samples collected in 2018. Arrows in (e) show some rings. Arrow in (f) shows the mesh texture of the polysaccharidic envelope of the rings.

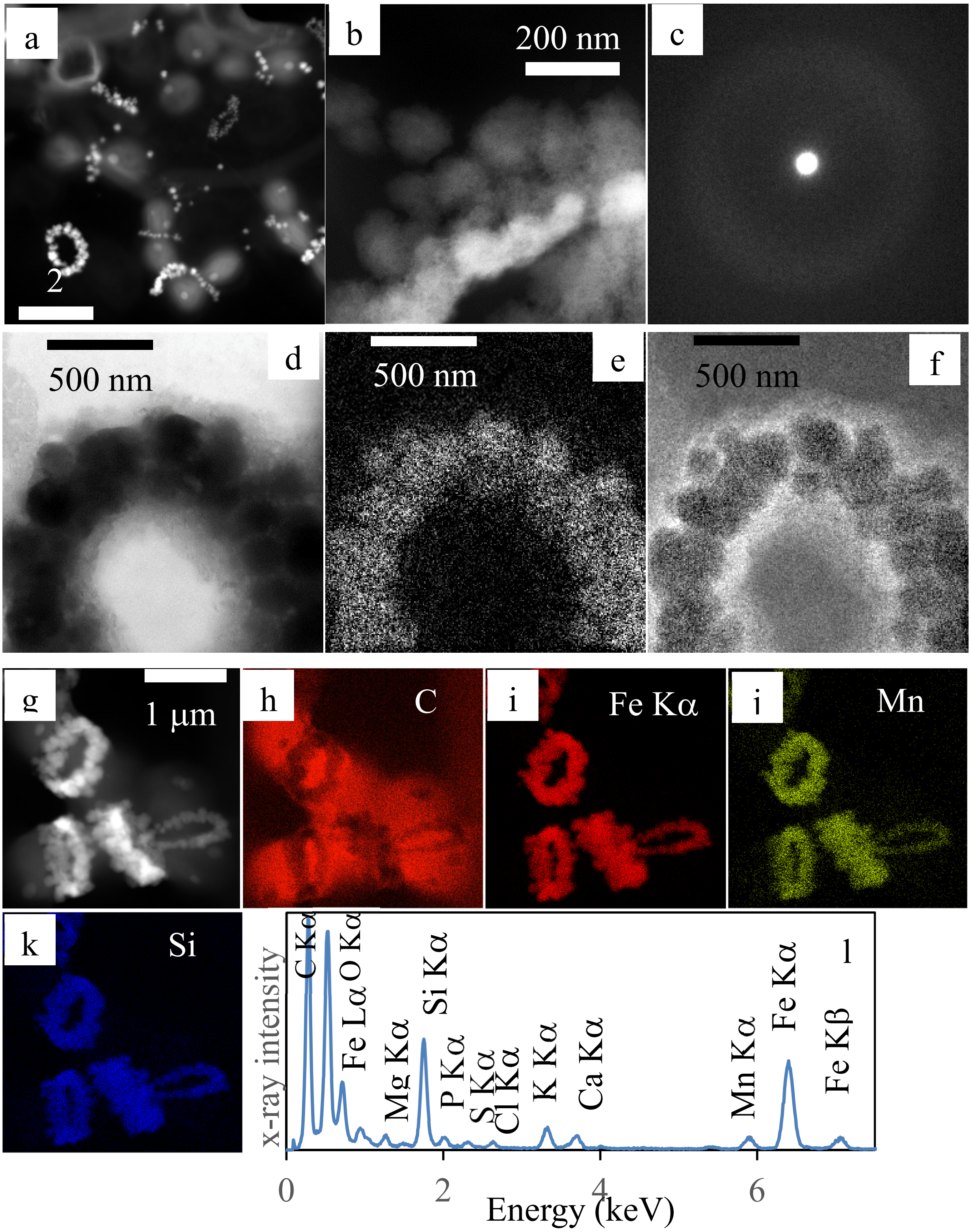
a
2
b
200 nm
c
d
500 nm
e
500 nm
f
500 nm
g
1 μm
h
C
i
Fe Kα
i
Mn
k
Si
l
x-ray intensity
C Kα
O Kα
Fe Lα
Mg Kα
Si Kα
P Kα
S Kα
Cl Kα
K Kα
Ca Kα
Mn Kα
Fe Kα
Fe Kβ
0
2
4
6
Energy (keV)

**Figure 2.** Transmission electron microscopy analyses of bacterial (Fe,Mn, Si)-rich rings. (a) Scanning transmission electron microscopy image in high-angular annular dark field (STEM-HAADF) mode of dividing cells collected in 2019. Rings which are composed of high atomic number elements appear in bright. (b) STEM-HAADF image at higher magnification of a ring collected in 2018, showing that it is composed of globules, which themselves show a fibrous texture. (c) Selected area electron diffraction pattern of a ring, characteristic of an amorphous material. (d) Zero-loss energy-filtered transmission electron microscopy (EFTEM) image of a ring collected in 2018. (e) EFTEM image at the Fe $L_{2,3}$ edges. (f) EFTEM image at the C K-edge. (g) STEM-HAADF image of cells with rings chemically mapped using energy dispersive x-ray spectrometry (EDXS). (h) Chemical map of carbon based on the C K$\alpha$ emission line. (i) Chemical map of iron based on the Fe K$\alpha$ emission line. (j) Chemical map of manganese based on the Mn K$\alpha$ emission line. (k) Chemical map of silicon based on the Si K$\alpha$ emission line. (l) EDXS spectrum of the rings observed in the chemical maps.

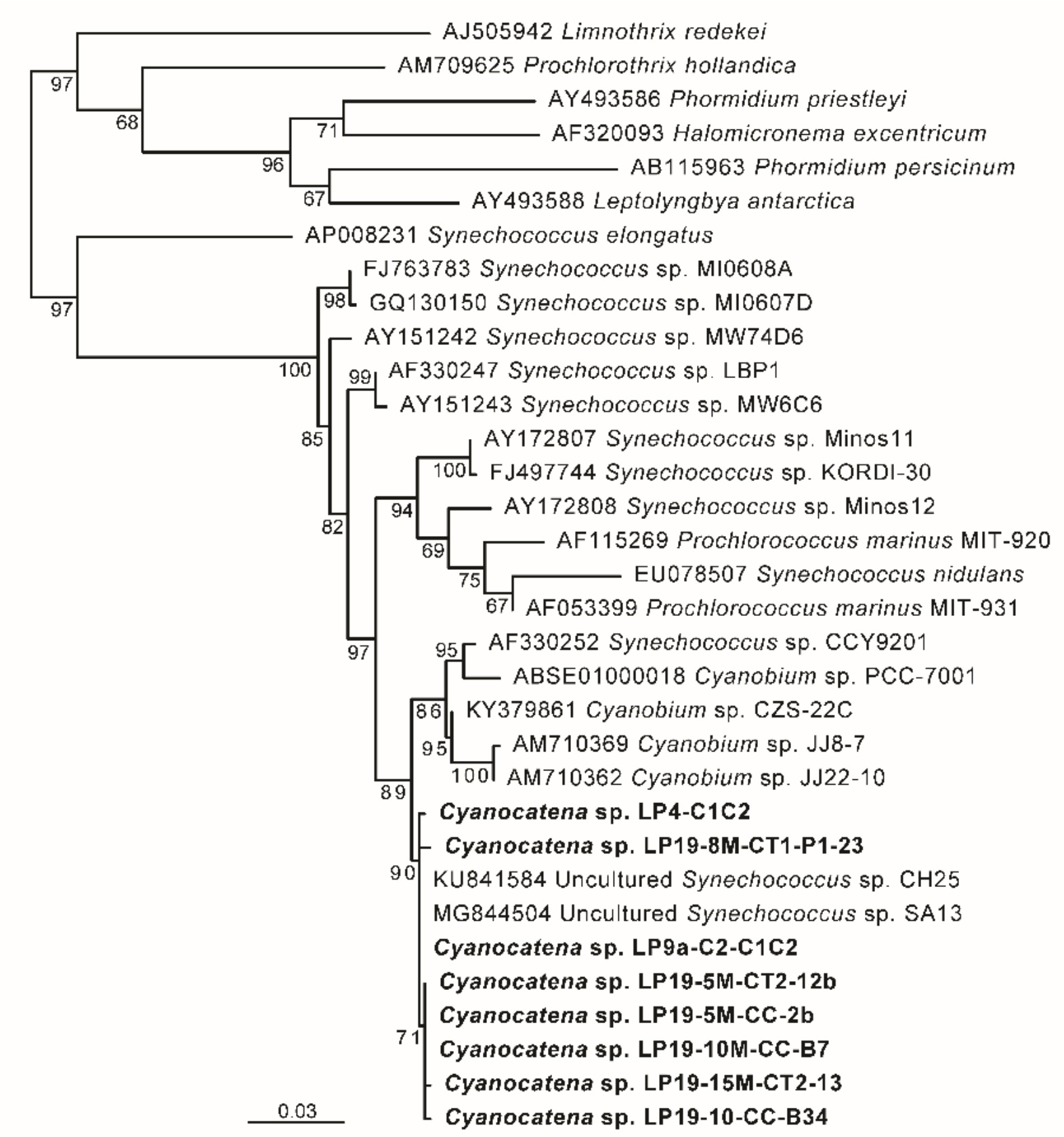

**Figure 3.** Maximum likelihood phylogenetic tree of cyanobacterial 16S rRNA gene sequences showing the position of the cells carrying (Fe,Mn)-rings. Numbers on branches are bootstrap proportions. The bolded sequences are from this study, derived from the cells sorted by flow cytometry and forming rings, and they were named *Cyanocatena* as explained in the discussion, section 4.2.

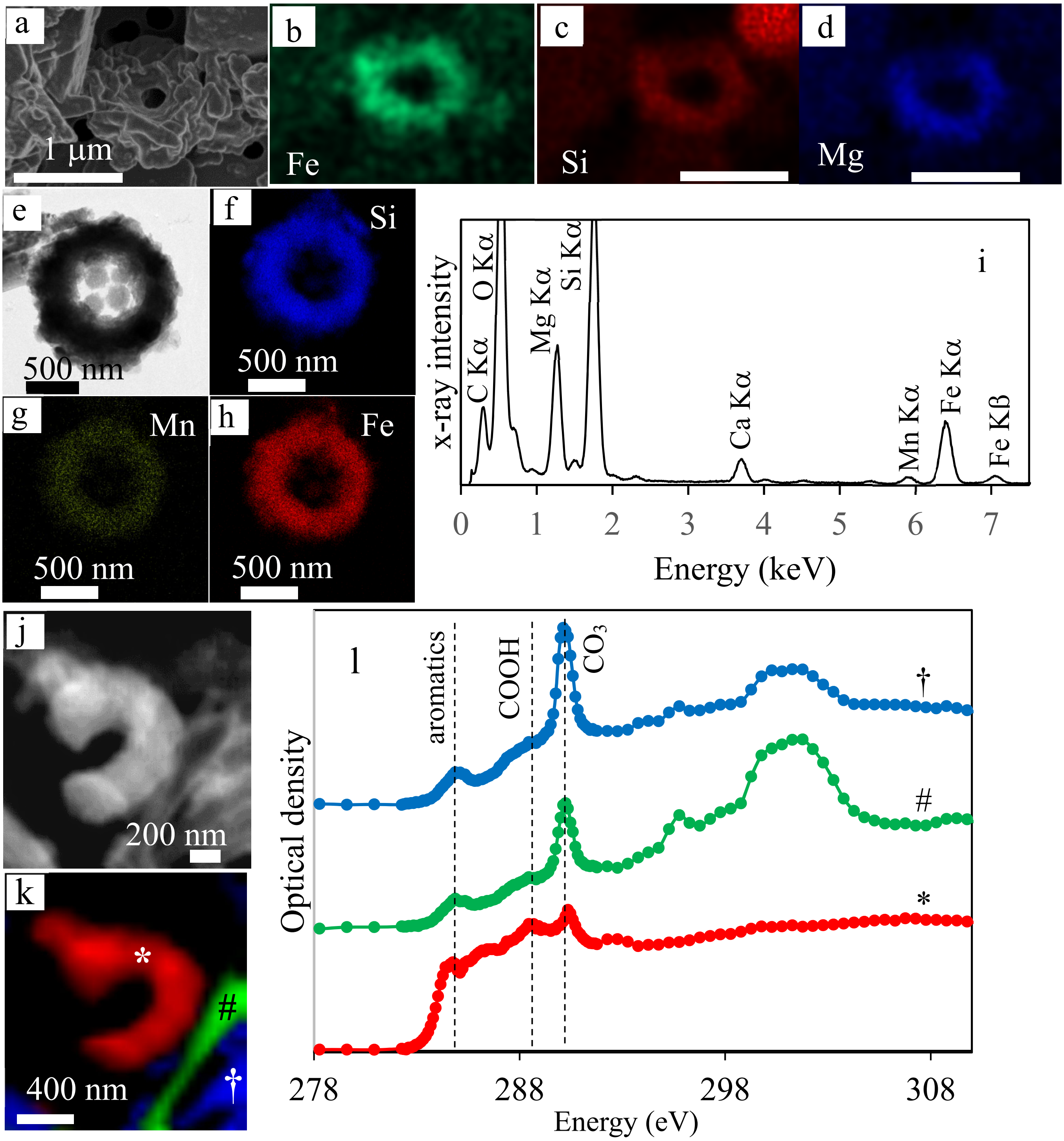

**Figure 4.** SEM, TEM and STXM analyses of cyanobacterial (Fe, Mn)-rings in Lake La Preciosa sediments. (a-d): SEM image in the secondary electron detection mode and EDXS chemical maps of Fe, Si and Mg of a ring in pristine (non-acid-leached)

sediments collected in 2016 at a 3 cm depth in the sediment core. Note that the surface of the rings seems to be covered by a smooth and folded layer in SE mode, suggesting that they are covered by organics. (e) Bright-field STEM image of a ring in acid-leached sediments. (f-h) Corresponding STEM-EDXS chemical maps of Si, Mn and Fe. (i) STEM-EDXS spectrum measured on the ring. (j) Carbon map obtained by subtracting an OD-converted STXM image aquired at 282 eV from an OD-converted image at 288.5 eV. (k-l) Spectral analysis at the C K-edge of the image shown in (j). (k): map of three carbon-containing compounds; carbon associated with rings is marked with a *; carbon typically in a carbonate grain is marked with a #; carbon in a carbonate functional group but with a different crystallographic orientation is marked with a †. (l) Corresponding spectra at the C K-edge (see symbols above the spectra). Vertical dashed lines are at 284.8 eV (attributed to aromatic functional groups), 288.5 eV (attributed to carboxylics, COOH, functional groups), 290.3 eV (attributed to carbonate, $CO_3$, groups). The ring analyzed by STXM was recovered from an acid-leached sediment sample at 1 cm depth in the sediment core (see Fig. SI-11 for TEM analyses).