# Peer review of "Biomineralization of amorphous Fe-, Mn- and Si-rich mineral phases by cyanobacteria under oxic and alkaline conditions"

_Biogeosciences, 2023_

## Author Response (AR1)

**Response to reviewers**

**Referee #1**

**General comments**

*The article reports the novel finding of Fe-Mn-Si rings formed by cyanobacteria in alkaline lakes under oxic conditions. Detailed characterization of the rings and the cyanobacterial species responsible for the biominerals are presented. The implications are tied to an intriguing re-thinking of the significance of early Earth Fe-Mn deposits as indicators of anoxic condition. Overall, I recommend publication after minor revision.*

 **Detailed comments**

- *A short explanation of the term "picocyanobacteria" would be useful for readers.*
  **We now provide some explanation of what picocyanobacterial are in the abstract.**

- *Line 25-30: "partially preserved" instead of "preserved"? Because evidences of degradation are visible.*
  **Done**

- *Line 196: can you specify how exactly you estimated the cell abundance?*
  **Cell abundance has been estimated by counting cells with SEM. This is now detailed in the material and method section.**

- *Line 209: is this sentence accurate? From the images, it looks like Mn is indeed variable, but much more abundant than Mg, Ca and K. The title also suggests a lot of Mn. I recommend re-wording to "contained ... variable amount of Mn, with lower amounts of Ca, K and Mg…"*

  **Yes, the sentence is accurate but we reformulated as suggested by the reviewer. And since it is difficult to tell the content of an element from images, defined based on ROI (they are meant to show the spatial distribution of the elements), we now refer to table SI-1, this is much clearer indeed for quantification.**

- *Line 216: it will be helpful to state here that samples pre-imaged by TEM seems to contain reduced Mn. It will be good for other researchers to know that TEM can produce artefacts.*
  **Right, we have now added this comment in the revised manuscript.**

- *Line 231: it was a bit confusing to read this sentence in combination with Figure 3, because there was no explanation of Cyanocatena beforehand. I recommend specifying in the figure caption that the bolded sequences are from this study, and that they were classified as Cyanocatena (refer to Discussion section xxx).*
  **Thank you. We now specify this in the caption of Figure 3.**

- *Do you have dissolved concentrations of Mn with depth in the water column?*

**Only in 2019 but at this date Mn was also below the detection limit for other depths. This is now specified in the revised manuscript.**

- *Figures, especially in the supplementary, can be made smaller so that the caption can be on the same page. It's a little bit frustrating to have to keep scrolling between pages to look at the caption and the figure, which can span a few pages long.*

**We have now modified the sizes of the figures in the supplementary so that all appear on the same page as the caption except for figure 2, the caption of which we moved to the front for reader's comfort.**

**Referee #2**

*Benzerara et al. report a novel biomineralization process that appears to be a controlled biomineralization of Fe-Mn-Si coprecipitates by certain picocyanobacteria. Their observations and data are compelling and high-quality, and this paper will ultimately be a great contribution to Biogeosciences. However, I recommend major revisions to the text and interpretation. I outline my substantial concerns below about their characterization of the precipitate as a silicate and issues with the writing flow and structure especially in the Intro and Discussion.*

**Major comments**

- *Abrupt transition from Fe/Mn cycles to Mexican crater lake. L44-45 sets up crater lake better but comes in middle of large Intro paragraph. Suggest re-ordering Intro to better set up investigation, and actually introduce Lake La Preciosa as the alkaline Mexican crater lake with an oxic water column. Why did the authors study this lake to start with? It seems like the silicate rings were a fortuitous find or was the lake being studied for its metal cycling for another reason?*

  **We completely agree. Therefore, we have now added sentences to the introduction to better explain that this finding was indeed fortuitous but also why we were interested by the study of this lake.**

- *Could the average or range of at least Fe, Si, and Mn content of these rings please be given in Fig. 2, a table, and/or in the text around L209? The authors need numbers to back up their interpretation of a Fe-Mn silicate.*

  **Of course. This was also suggested by reviewer #1 and this was an oversight in the first submission. We now provide numbers under brackets and refer to Table SI-1 which provides those numbers.**

- *I'm actually not convinced that these precipitates can be called silicates. They are apparently coprecipitates of Fe and Si but seems like they could be entirely amorphous or a SiO2 gel with bound Fe(III) and other elements, or a ferrihydrite with adsorbed silica and other elements? See, for example, Dyer et al 2012 (Insights into the crystal*

*and aggregate structure of Fe3+ oxide/silica coprecipitates in American Mineralogist) and Emmanuel Doelsch's work on Fe(III) crystallized in the presence of silica (2000 & 2001 in Langmuir and 2003 in Colloids and Surfaces A: Physicochem. Eng. Aspects). Additionally, Fig. S3 shows the pristine ring matches quite well with ferrihydrite in the Fe L2,3 edge. I'm not sure this transition can distinguish coordination environment as well as redox state, but Bourdelle et al's (in which Benzerara is 2nd author) measurements of a fully ferric silicate appears less similar to the ring than ferrihydrite.*

**We thank the reviewer for this comment as we realize that eventually, modifying the denomination of the phase reinforces the main point of this paper, i.e. that the rings are primarily Fe- and Mn-rich. We now mention that this phase could alternatively be a mixture of Fe and Mn oxyhydroxides with sorbed silica and that the nature of the exact local structure of this phase should be determined in the future. As inferred by the reviewer Fe $L_{2,3}$ edges do not really help in assessing how Si tetrahedra and Fe and Mn are linked together as they are primarily more sensitive to redox and some symmetry parameters by contrast with the K-edge. And we could not perform, e.g., EXAFS which could have been key in deciphering the local order of these amorphous compounds, since we are dealing with complex mixtures of various phases and the rings are only part of these. So most of our suggestions about their possible nature relies on their stoichiometry that is more reminiscent of silicates. Yet, whether these Fe and Mn-rich phases are silicates or not is not a key point of the study. We agree with the reviewer that what we know for sure is that they are Fe, Mn and Si-rich, that they are amorphous, and that the silicate character remains to be ascertained and at this point is a speculation. At least, we can suggest that these phases are potential precursors to silicate phases. Overall, and according to the reviewer's comment, we have now revised the manuscript to use a more conservative denomination of this phase by avoiding the term silicate and using instead 'amorphous Fe/Mn-rich phases' or 'Fe-, Mn- and Si-rich phases' and mention they could be a potential precursor to Fe/Mn silicates.**

- *Are the Fe-Si rings actually well-preserved? I agree with Ref 1 that they is evidence of degradation of the rings in the sediments and add that it looks like Mg replaces Mn (or some of the Fe?) in Figs. 4 and S10 (Mn isn't even shown on Fig. S10). This replacement should be noted in the text.*

**This is absolutely right. In the first version, we meant a morphological preservation and the fact that they were still Fe-rich. We agree that based on our available data, the chemical composition of the sediment rings seems a bit different from the ones in the water column, with more Mg and less Mn (although there is some variability in the water column rings) and we now clearly specify this and provide a semi quantitative analysis in Fig S11 and also note in the caption of Fig S10 that Mn was hardly detected by SEM-EDXS. We believe that only a thorough analysis of sediment rings with a particular care about a potential depth evolution (in the water column and/or in the sediments) will allow being more specific about the potential transformations experienced by the rings.**

- *The authors touch on possible precipitation mechanisms in L306-313 but it would be nice for the authors to expand on hypotheses for the mechanism of the Fe,Mn-Si coprecipitate formation. For example, is cellular division known to cause a local region of higher or lower pH that might promote Fe-Si (+ other elements) precipitation? Or could cell division release of Fe2+ that could immediately precipitate as ferric oxides and adsorb Si? The observation of the extracellular compartment or envelope is fascinating but isn't returned to in the brief text exploring underlying formation process. Could the Mn derive from intracellular stores as well? Potential mechanisms are then further explored in the next section, especially L332-342. Upon revision, the authors need to consolidate all their text on what might drive the formation of these solid rings, perhaps as a focused section in the Discussion that takes the reader from cells to precipitation. I'd suggest this should come after the discussion of other picocyanobacterial that form iron oxide rings, because that is really helpful context.*

  **Following the reviewer's suggestions, we restructured the discussion. We first inverted the two sub-sections so that the discussion about other picocyanobacterial forming iron oxides comes first. Then we grouped the different speculations about how the rings may form. We also added all suggestions by the reviewer about possible additional mechanisms involved. In the end, we specify that only future work on an isolated strain andor the genome of the bacterium may help proceed further on this topic.**

**Minor comments**

- *L92 no need for 'Besides'*
  **'Besides' was removed**

- *L85 Missing italics on bacterial name*
  **The name has been italicized**

- *It'd be helpful to explicitly make the connection between carboxylic groups identified in Supp Fig 8 and a polysacchardic composition of the envelope*
  **We know explain more explicitly the connection between the carboxylic groups in Fig S8 and the polysaccharidic composition of the envelope on Lines 267-269.**

- *In all the STXM spectra with dashed line labels, it would be really helpful to add a bit of text to identify the functional groups within the figure, not just the caption.*
  **Figure 4 and S8 have been modified accordingly.**

- *L285 should have 'or' instead of 'and' between two mechanisms from sentence set-up*
  **Exact. We removed either to keep an and/or in the sentence as these two mechanisms may play a role at the same time**

- *L304 should give ratio of Fe+Mn+Mg/Si+Al for a smectite that fits with the observed ratios – I thought a typical smectite would have a lot more Si+Al than Mg+Fe+Mn (a ratio of ~0.5).*
  **We added the Fe/Si of ferrosaponite which is ~3/4**

- *L334-5 - See Lingappa et al.'s finding of large amounts of intracellular stores of Mn in cyanobacteria (PNAS 2021, 'An ecophysiological explanation for manganese enrichment in rock varnish').*
  **Thank you very much. This reference has been added as a source for an addition possible function of Mn accumulation.**

- *L359-360 – The formation of Fe-Mn-rich phases in an oxic water column is certainly intriguing, but have the authors measured the concentration of Fe and Mn in the sediments? Do these rings actually produce a sedimentary enrichment in Fe & Mn? The authors could also consider comparing to Fe-Mn nodules and crusts on the seafloor, which are also formed under oxic conditions. See Hein and Koschinsky 2014 (in Treatise on Geochemistry 2$^{nd}$ edition).*
  **Yes. These numbers are now provided in the revised version for both Fe$_2$O$_3$ and MnO content of the sediments. In this case, the enrichment in sediments is not particularly high. Thank you for the recommended reference, we have now added one comment in the conclusions about Fe-Mn nodules which however seem to form under diagenetic conditions and therefore involve reduction/oxidation processes.**